# Enhancing Retinal Vessel Segmentation Generalization via Layout-Aware Generative Modelling

## Abstract

Generalization in medical segmentation models is challenging due to limited annotated datasets and imaging variability. To address this, we propose **R**etinal **L**ayout-**A**ware **D**iffusion (RLAD), a novel diffusion-based framework for generating controllable layout-aware images. RLAD conditions image generation on multiple key layout components extracted from real images, ensuring high structural fidelity while enabling diversity in other components. Applied to retinal fundus imaging, we augmented the training datasets by synthesizing paired retinal images and vessel segmentations conditioned on extracted blood vessels from real images, while varying other layout components such as lesions and the optic disc. Experiments demonstrated that RLAD-generated data improved generalization in retinal vessel segmentation by up to 8.1%. Furthermore, we present **REYIA**, a comprehensive dataset comprising 585 manually segmented retinal images. We make the REYIA dataset and our source code open (upon publication) ⬡.

## 1  Introduction

Deep learning has achieved remarkable success across various domains, but its progress often depends on access to large annotated datasets. In fields such as natural language processing, vision-language modeling, and image generation, synthetic data from large models has driven significant advancements [1–6]. However, in medical imaging, particularly retinal vessel segmentation, data scarcity and variability in imaging conditions remain persistent limitations [7–10]. Retinal vessel segmentation is critical for the diagnosis of ocular and systemic diseases [11–14], yet the creation of annotated datasets demands a considerable amount of time, specialized expertise, and consistency across imaging devices [15].

Retinal vessel segmentation involves two tasks: general vessel segmentation, which identifies the vasculature, and artery/vein (AV) segmentation, which also differentiates arteries from veins. This distinction provides insights into vessel-specific pathologies[16, 17]. However, AV segmentation requires complex annotations, making it challenging to obtain sufficient labeled data for robust training.

Generative models like GANs and VAEs have been explored to address data scarcity in medical imaging [18, 19]. When applied to retinal images, these models often encounter challenges, including difficulties in preserving anatomical fidelity and issues with training stability [20]. Diffusion models have recently emerged as powerful tools for generating diverse high-fidelity images, with superior stability and detail preservation, compared to GANs and VAEs [21, 22]. Despite their success in image synthesis tasks across domains, e.g., natural image generation and text-to-image modeling, their application in medical imaging has largely focused on generating synthetic images rather than directly enhancing segmentation performance through data augmentation.

Submitted to 39th Conference on Neural Information Processing Systems (NeurIPS 2025). Do not distribute.

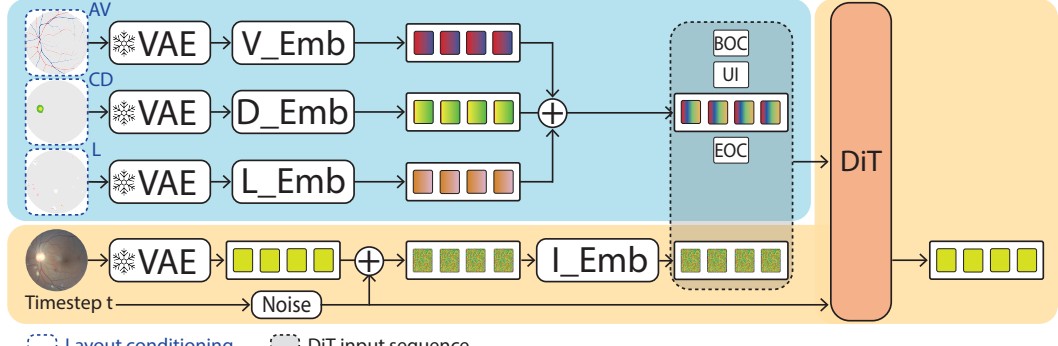

Figure 1: **RLAD Architecture.** The original fundus image and segmentation maps for artery/vein (AV), the optic cup/disc (CD), and lesions (L) are encoded into latent representations using a frozen VAE. Gaussian noise is added to the image latent, and each latent (image, CD, AV, and L) is projected into the DiT [25] input space via distinct projections. Condition embeddings for AV, CD, and L are summed into a single embedding, $c$. The DiT input consists of a beginning-of-conditioning (BOC) token, user input (UI), $c$, an end-of-conditioning (EOC) token, and the noised image latent. The DiT outputs the corresponding denoised image latent. The UI token specifies whether a layout component is guided by user input or defaults to a neutral embedding when absent.

To address these limitations, we propose Retinal Layout-Aware Diffusion (**RLAD**), a diffusion-based framework for the controllable generation of synthetic retinal images. By conditioning on multiple key retinal structures—such as artery/vein (AV), the optic cup/disc (CD), and lesions (L)—RLAD preserves essential vascular layouts while introducing variability in other regions. This enables the creation of paired image-segmentation maps that expand training datasets without compromising structural integrity. Synthetic data generated by RLAD improve segmentation model robustness across diverse imaging conditions and acquisition settings.

We evaluated RLAD-generated data using state-of-the-art visual encoders such as Vision Transformers [23] and Swin Transformers [24], and demonstrate consistent improvements in generalization performance under distribution shifts (up to 8.1%). Additionally, we introduce **REYIA**, the largest multi-source collection of 585 retinal images with human reference AV segmentation, which not only complements our synthetic data but also demonstrates strong baseline performance, further validating the effectiveness of our synthetic data. In summary, the main contributions of this work are:

- A novel multi-layout-aware generative model (**RLAD**) that synthesizes diverse yet anatomically accurate retinal images while preserving semantic structures.

- Demonstrating consistent segmentation performance improvements across state-of-the-art architectures using RLAD-generated data.

- Introducing **REYIA**, the largest multi-source collection of datasets for AV-segmented retinal fundus images.

## 2  Related Work

Retinal AV segmentation plays a critical role in diagnosing microvascular pathologies [26-30]. Early methods [8, 31–34], such as Little W-Net [7], focused on compact convolutional neural networks to reduce computational complexity. More recently, LUNet achieved state-of-the-art performance on optic disc-centered images but struggled to generalize to macula-centered images [9]. This underscores the primary challenge of achieving robust generalization across diverse retinal imaging conditions.

Generative adversarial networks have been extensively used for retinal image synthesis, often conditioning the generation process on features such as vessel or lesion masks [35, 36]. While these methods produced visually realistic images, they frequently lacked anatomical accuracy and robustness [20], limiting their effectiveness for downstream tasks like AV segmentation. To address these issues, Go et al. [20] proposed a hybrid approach that combined a diffusion model for generating AV masks with a conditional GAN for synthesizing retinal images. Their method preserved patient privacy and demonstrated that synthetic images could lead to AV segmentation performance comparable to models trained on real data. However, it failed to further enhance AV segmentation performance

further, possibly due to limited variability in the generated AV masks, which may have propagated to the synthesized images.

Diffusion models have demonstrated remarkable generative capabilities across various domains, including image synthesis, video generation, layout and 3D modeling [1, 21, 37–43]. Recent advancements, such as classifier-free guidance [44] enable precise control over conditioning signals during generation, making these models well-suited for structured image synthesis tasks. Transformer-based architectures such as DiT [25] further enhance performance by capturing long-range dependencies.

Building on these developments, we propose a multi-layout-aware diffusion framework specifically designed for retinal fundus image synthesis. Unlike prior approaches, our method conditions generation on multiple retinal layout components —AV, CD, and L—extracted from real, non-annotated images using pretrained segmentation models. This minimizes error propagation and enhances realism while addressing domain generalization challenges in AV segmentation tasks through synthetic data augmentation.

# 3 Datasets

This section introduces the new datasets created for this study and provides an overview of the datasets used for diffusion model training and downstream segmentation tasks. For additional details, please refer to the appendix.

## 3.1 New Datasets

We introduce REYIA, a curated set of 585 retinal fundus images annotated with AV blood vessel segmentations using the open-access Lirot.ai software [15] and summarized in Table 1. To enhance diversity, REYIA includes manually segmented images as part of this research from nine datasets: FIVES [45], TREND [46], GRAPE [47], MESSIDOR [48], MAGRABIA [49], PAPILA [50], MBRSET [51] AV-WIDE [52] and ENRICH. ENRICH is a new dataset collected for this study, consisting of 111 retinal fundus images (IRB S60649). AV-WIDE, which initially contained only skeletonized vessels, was reannotated to include complete vessel segmentations.

| Dataset | # Samples | Image Center | FOV ($^\circ$) | Region | Resolution (px) |
|---|---|---|---|---|---|
| GRAPE$^\dagger$ [47] | 81 | M | 50 | China | 1444x1444 |
| MESSIDOR$^\dagger$ [48] | 67 | M | 45 | France | 1444x1444 |
| PAPILA$^\dagger$ [50] | 78 | D | 30 | Spain | 1444x1444 |
| MAGHREBIA$^\dagger$ [49] | 69 | M, D | 30 | Maghreb | 1444x1444 |
| ENRICH$^*$ | 111 | D | 45 | Belgium | 1958x2196 |
| FIVES$^\dagger$ [45] | 75 | M | 45 | China | 1444x1444 |
| AV-WIDE$^\dagger$ [52] | 26 | D | Ultra wide | USA | 829x1531 |
| TREND $^\dagger$ [46] | 48 | M | 30 | Montenegro | 2560x2560 |
| MBRSET$^\dagger$ [51] | 30 | M | 30 | Brazil | 1444x1444 |

Table 1: **REYIA datasets collection** released with this work. Datasets marked with $^\dagger$ were annotated specifically for this work, and those marked with $^*$ were both introduced and annotated here.

## 3.2 Diffusion Model Datasets

To train RLAD, we curated 112,320 retinal fundus images from publicly available datasets spanning diverse imaging conditions, fields of view (FOV), and pathologies. The sources include widely used datasets: UZLF [53], GRAPE [47], MESSIDOR [48], PAPILA [50], MAGRABIA [49], ENRICH, 1000images [54], DDR [55], EYEPACS [56], G1020 [57], IDRID [58] and ODIR [59]. Evaluation of the realism of the generated images, in comparison to real images, was performed on the DRTiD dataset [60].

## 3.3 AV Segmentation Datasets

### 3.3.1 Datasets for Segmentation Model Training

To train our segmentation models, we constructed a composite dataset combining the UZLF dataset with newly annotated versions of GRAPE, MESSIDOR, ENRICH, MAGRABIA, and PAPILA. These datasets feature high-resolution retinal fundus images with FOVs ranging from 30° to 45° and encompass a variety of ophthalmic conditions and patient populations.

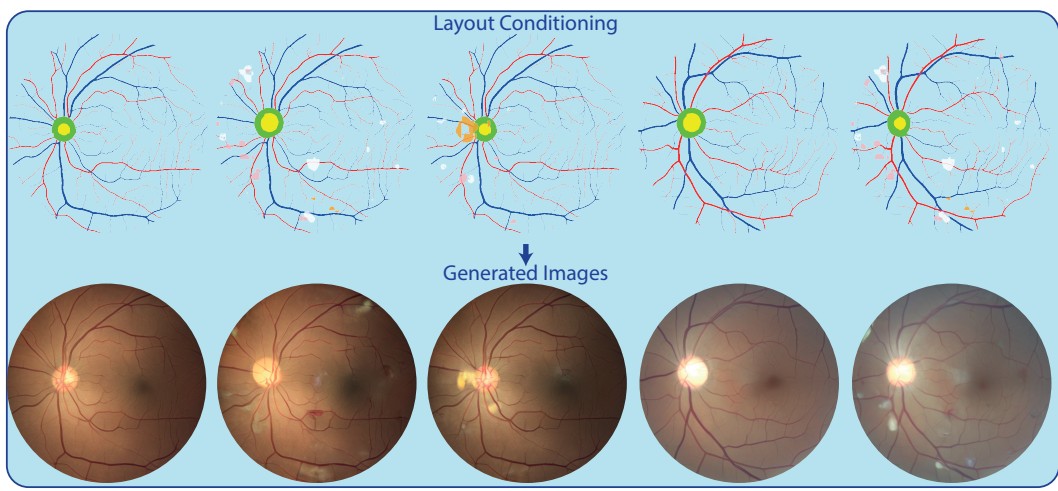

Figure 2: **Retinal Layout-Aware Diffusion Qualitative Examples.** Top: user-defined layout components inputs (artery/vein in red/blue, optic disc/cup in green/yellow, and lesions in white/pink/orange). Bottom: corresponding generated fundus images.

### 3.3.2 Datasets for Segmentation Model Evaluation

To assess generalization performance under varying levels of distribution shift, we evaluated our segmentation models across three categories of datasets:

**In-Domain (Local):** Data collected from the same hospital under similar acquisition conditions to those as one of the training datasets, ensuring minimal distribution shifts.

**Near-Domain (External):** Data from different hospitals and environment, introducing moderate distribution shifts. This category includes HRF [61], INSPIRE [9, 62], UNAF [9, 63] and the reannotated FIVES dataset.

**Out-of-Domain (OOD):** Data that significantly differ from the training distribution, used to evaluate the model robustness across diverse imaging conditions. It includes AV-WIDE for ultra-wide-angle images, IOSTAR [64] for laser-based images, DRIVE [65, 66] for low-resolution images, RVD [10] for video frames from handheld devices, TREND and MBRSET for handheld device images.

## 4 Method

Our objective is to generate realistic retinal images based on key retinal layout components, specifically AV, CD, and L, extracted from real retinal fundus images.

### 4.1 Layout Extraction

We extract retinal layouts using open-source models for L segmentation [67] and CD segmentation [17, 68]. For AV segmentation, we retrained a SwinV2$_{\text{tiny}}$-based model on our annotated datasets with data augmentation techniques such as random color jitter, flips, and rotations. These extracted retinal layout components serve as input to the diffusion process. The impact of the layout extractor used is further discussed in the appendix.

### 4.2 Retinal Layout-Aware Diffusion

Our approach builds upon latent diffusion [69] and DiT [25]. The forward diffusion process [21, 37] gradually adds Gaussian noise to an image $x_0$, producing $x_t$. This process is defined as:

$$q(x_t \mid x_0) = \mathcal{N}(x_t; \sqrt{\overline{\alpha}_t}x_0, (1 - \overline{\alpha}_t)I), \tag{1}$$

where the noise schedule $\{\overline{\alpha}_t\}$ follows a linear strategy as explored in [21]. The reverse process approximates the denoising steps to reconstruct $x_0$:

$$p_\theta(x_{t-1} \mid x_t, c) = \mathcal{N}(x_{t-1}; \mu_\theta(x_t, c), \Sigma_\theta(x_t, c)), \tag{2}$$

where $c$ denotes conditioning information. Instead of operating directly in pixel space, we adopt latent diffusion and perform these operations in a compressed latent space of a frozen VAE. This

allows us to refine latent representations $z_t$ iteratively towards $z_0$, improving computational efficiency and scalability.

To incorporate conditional information into the diffusion process, we extract the layout components (AV, CD and L) from the input data. These components are embedded into the transformer's latent space using dedicated projection heads: $V_{\text{emb}}$, $D_{\text{emb}}$ and $L_{\text{emb}}$.

$$c_{\text{AV}} = V_{\text{emb}}(\text{AV}), \quad c_{\text{CD}} = D_{\text{emb}}(\text{CD}), \quad c_{\text{L}} = L_{\text{emb}}(\text{L}).$$

To handle both fully and partially conditional cases, we used user input (UI) tokens. Each token indicates whether a component is user-defined (guided) or neutral (unconditional). During training, each layout component is either provided or masked with a certain probability, allowing the model to learn both conditional and unconditional scenarios. This probabilistic masking is applied independently to each component. When a component is masked, it is replaced with a "black" image embedding, and its corresponding UI token is updated to signal the absence of guidance:

$$\text{UI} = [\text{UI}_{\text{AV}}, \text{UI}_{\text{CD}}, \text{UI}_{\text{L}}],$$

allowing flexible control over the conditioning process. The final conditioning vector is computed as:

$$c = c_{\text{AV}} + c_{\text{CD}} + c_{\text{L}}.$$

which is fed into the transformer as part of a sequence:

$$[\text{BOC}, \text{UI}, c, \text{EOC}, z_t],$$

where BOC and EOC mark the beginning and end of the conditioning tokens, respectively. After the transformer processes this sequence, only the image tokens are retained to produce $z_{t-1}$. This design ensures that conditioning signals guide the denoising process without remaining entangled in the final latent representation. A schematic overview of our architecture is provided in Figure 1.

**Training Objective.** Following DDPM [21], we adopt a noise prediction loss. Instead of directly modeling $\mu_\theta$ and $\Sigma_\theta$, our model predicts the noise $\epsilon$ added at a randomly chosen timestep $t$:

$$L_{\text{simple}} = \mathbb{E}_{z_0, t, \epsilon}\left[\|\epsilon - \hat{\epsilon}_\theta(z_t, t, c)\|^2\right]. \tag{3}$$

Minimizing this MSE loss enables the model to accurately denoise latent representations, effectively learning to reverse the diffusion process. By incorporating tokens that differentiate between user-defined and neutral embeddings for each layout component, the model can both generate anatomically guided images when specific conditions are provided, and produce diverse, unconstrained samples in the absence of such guidance. This flexibility ensures that the model adapts seamlessly to varying levels of conditional input, balancing anatomical fidelity with generative diversity.

**Sampling.** To generate new images, we start from a random Gaussian latent $z_T \sim \mathcal{N}(0, I)$ and iteratively remove noise at each diffusion step $t$. Our model predicts the added noise $\hat{\epsilon}_\theta(z_t, t, c)$, where $c$ includes tokens for AV, CD, and L layouts.

We employ classifier-free guidance [44] to control how closely the model adheres to provided conditions. At each step, two predictions are made: one conditional ($c$) and one unconditional ($c = \emptyset$). These are combined as:

$$\hat{\epsilon}_\theta^{\text{guided}}(z_t, t, c) = \hat{\epsilon}_\theta(z_t, t, \emptyset) + w\left(\hat{\epsilon}_\theta(z_t, t, c) - \hat{\epsilon}_\theta(z_t, t, \emptyset)\right), \tag{4}$$

where $w$ is a guidance scale. Higher $w$ yields more faithful adherence to the conditions, lower $w$ allows more diversity.

By iteratively applying guided noise predictions until reaching $z_0$, we decode $z_0$ using the VAE to produces a synthetic retinal fundus image. This approach balances anatomical fidelity when conditions are provided with greater diversity when they are neutral or absent. Examples of generated images are shown in Figure 2.

## 4.3 Backbone Pretraining

We investigate pretraining strategies to enhance segmentation performance, focusing on two key approaches: Masked Autoencoders (MAE) [74] and Windowed Contrastive Learning (WCL) [75].

| Backbone | Local | | External | | | | OOD | | | | | | Average | |
|---|---|---|---|---|---|---|---|---|---|---|---|---|---|---|
| | UZLF | LES-AV | HRF | INSPIRE | FIVES | UNAF | AV-WIDE | IOSTAR | DRIVE | RVD | TREND | MBRSET | External | OOD |
| RMHAS[8] | - | 60.0 | 48.0 | - | - | - | - | 55.0 | 60.0 | - | - | - | - | - |
| RVD$_{\text{Swin-L}}$ [10] | - | - | - | - | - | - | - | - | 57.3 | 53.0 | - | - | - | - |
| Little W-Net [7] | 80.7 | 82.0 | 58.1 | 71.3 | 73.5 | 68.6 | 43.1 | 29.9 | 61.3 | 34.7 | 53.4 | 50.4 | 67.9 | 45.5 |
| Automorph [34] | 76.3 | 84.0† | 77.4† | 71.1 | 72.5 | 65.9 | 50.1 | 54.9 | 78.1† | 34.1 | 66.6 | 63.7 | 71.7† | 57.9† |
| VascX [70] | 80.6 | 81.8 | 75.6 | 74.9 | 80.4 | 73.1 | 49.8 | 52.1 | 73.6 | 42.6 | 71.9 | 73.2 | 76.0 | 60.5 |
| LUNet [9] | 83.2 | 83.5 | 73.1 | 75.5 | 86.0 | 74.4 | 69.3 | 56.7 | 71.1 | 35.2 | 71.1 | 63.2 | 77.3 | 61.1 |
| DinoV2$_{\text{small}}$ [71] | 81.6$_{\pm0.9}$ | 82.4$_{\pm1.4}$ | 74.2$_{\pm0.8}$ | 76.6$_{\pm0.9}$ | 82.7$_{\pm1.0}$ | 72.9$_{\pm1.9}$ | 59.4$_{\pm2.4}$ | 57.2$_{\pm2.7}$ | 75.0$_{\pm1.2}$ | 45.4$_{\pm0.6}$ | 67.1$_{\pm1.5}$ | 79.6$_{\pm1.1}$ | 76.6 | 64.0 |
| + RLAD (Our) | 81.8$_{\pm0.9}$ | 82.8$_{\pm1.3}$ | 75.1$_{\pm0.8}$ | 77.5$_{\pm0.7}$ | 83.6$_{\pm1.0}$ | 73.7$_{\pm1.8}$ | 58.3$_{\pm2.1}$ | 65.3$_{\pm3.2}$ | 76.8$_{\pm1.1}$ | 46.7$_{\pm0.6}$ | 70.8$_{\pm1.5}$ | 81.9$_{\pm1.5}$ | 77.5 | 66.6 |
| Δ | +0.2 | +0.4 | +0.9 | +0.9 | +1.1 | +0.8 | -1.1 | +8.1 | +1.8 | +1.3 | +3.7 | +2.3 | +0.9 | +2.6 |
| RETFound [72] | 81.2$_{\pm1.0}$ | 82.3$_{\pm1.5}$ | 77.7$_{\pm1.1}$ | 75.8$_{\pm0.9}$ | 82.1$_{\pm1.0}$ | 71.8$_{\pm1.9}$ | 63.2$_{\pm1.9}$ | 63.0$_{\pm3.3}$ | 75.1$_{\pm1.2}$ | 42.5$_{\pm0.8}$ | 70.1$_{\pm1.4}$ | 78.4$_{\pm1.7}$ | 76.9 | 65.2 |
| + RLAD (Our) | 83.1$_{\pm1.0}$ | 83.6$_{\pm1.5}$ | 80.2$_{\pm1.6}$ | 78.4$_{\pm1.0}$ | 86.3$_{\pm0.9}$ | 74.6$_{\pm1.9}$ | 69.5$_{\pm1.8}$ | 70.5$_{\pm3.0}$ | 77.1$_{\pm1.2}$ | 46.4$_{\pm0.8}$ | 76.9$_{\pm1.4}$ | 79.1$_{\pm1.7}$ | 79.9 | 69.9 |
| Δ | +0.9 | +1.3 | +2.5 | +2.6 | +4.2 | +2.8 | +6.3 | +7.5 | +2.0 | +3.9 | +6.8 | +0.7 | +3.0 | +4.7 |
| SwinV2$_{\text{tiny}}$ [73] | 82.8$_{\pm0.8}$ | 83.4$_{\pm1.4}$ | 79.9$_{\pm1.4}$ | 78.1$_{\pm0.9}$ | 85.9$_{\pm0.8}$ | 74.3$_{\pm1.9}$ | 68.1$_{\pm2.0}$ | 67.6$_{\pm3.1}$ | 76.0$_{\pm1.1}$ | 44.1$_{\pm0.8}$ | 76.2$_{\pm1.4}$ | 81.5$_{\pm2.7}$ | 79.6 | 68.9 |
| + RLAD (Our) | 83.0$_{\pm0.8}$ | 83.6$_{\pm1.4}$ | 80.2$_{\pm1.3}$ | 78.3$_{\pm0.9}$ | 86.3$_{\pm0.8}$ | 74.6$_{\pm1.9}$ | 69.5$_{\pm2.0}$ | 71.3$_{\pm2.7}$ | 77.1$_{\pm1.4}$ | 46.3$_{\pm0.7}$ | 77.1$_{\pm1.1}$ | 83.7$_{\pm2.0}$ | 79.9 | 70.8 |
| Δ | +0.2 | +0.2 | +0.3 | +0.2 | +0.4 | +0.3 | +1.4 | +3.7 | +1.1 | +2.2 | +1.1 | +2.0 | +0.3 | +1.9 |
| SwinV2$_{\text{large}}$ [73] | 83.2$_{\pm0.9}$ | 83.6$_{\pm1.4}$ | 80.4$_{\pm1.3}$ | 79.0$_{\pm0.9}$ | 87.2$_{\pm0.8}$ | 75.5$_{\pm1.7}$ | 70.9$_{\pm2.1}$ | 73.5$_{\pm3.1}$ | 76.5$_{\pm1.1}$ | 48.2$_{\pm0.7}$ | 77.4$_{\pm1.4}$ | 86.0$_{\pm1.6}$ | 80.5 | 72.1 |
| + RLAD (Our) | 83.2$_{\pm0.9}$ | 83.6$_{\pm1.5}$ | 80.4$_{\pm1.3}$ | 79.1$_{\pm0.9}$ | 87.3$_{\pm0.8}$ | 75.8$_{\pm1.7}$ | 71.2$_{\pm2.2}$ | 74.5$_{\pm2.8}$ | 77.1$_{\pm1.0}$ | 48.2$_{\pm0.7}$ | 77.6$_{\pm1.4}$ | 86.2$_{\pm1.6}$ | 80.7 | 72.5 |
| Δ | +0.0 | +0.0 | +0.0 | +0.1 | +0.1 | +0.3 | +0.3 | +1.0 | +0.6 | +0.0 | +0.2 | +0.2 | +0.2 | +0.4 |

Table 2: **RLAD Results.** Quantitative comparison of RLAD-generated data integrated into DinoV2, RETFound, and SwinV2 across model sizes. Baselines are trained on datasets from section 3.3. Evaluation spans Local, External, and OOD benchmarks, with average performance for External and OOD. Previous state-of-the-art performance (gray) reflects open-source inference or reported results. Performance is the average Dice score for artery and vein. † indicates data leakage during training.

MAE facilitates robust representation learning by reconstructing masked inputs, effectively teaching the model to predict missing portions of an image. WCL, initially designed for depth estimation, employs contrastive learning on small image patches while maintaining local spatial relationships, making it particularly suitable for semantic segmentation tasks. Furthermore, we explore multi-objective pretraining [76–78], by combining MAE and WCL to develop richer representations and improve downstream task performance. The dataset used for pretraining aligns with the one employed to train RLAD.

## 4.4 Enhancing AV Segmentation with RLAD

The synthetic images generated by RLAD serve as powerful data augmentation tools for vessel segmentation models. By preserving vascular structures while varying other characteristics (e.g., disc or lesions), these images enrich training datasets without requiring additional manual annotations.

Let a vessel segmentation model be denoted as $\mathcal{S}$, trained on real retinal images $x_{\text{orig}}$ with ground truth AV annotations $y$. The segmentation loss combines Dice loss and Binary Cross-Entropy (BCE) where $L^{\text{A}}$ and $L^{\text{V}}$ specifically represent the loss terms computed over artery and vein, respectively:

$$L_{\text{seg}} = 0.5 \cdot (L_{\text{Dice}}^{\text{A}} + L_{\text{BCE}}^{\text{A}}) + 0.5 \cdot (L_{\text{Dice}}^{\text{V}} + L_{\text{BCE}}^{\text{V}}). \tag{5}$$

The total training objective includes supervised loss on real images and consistency loss on synthetic images:

$$L_{\text{total}} = L_{\text{seg}}(\mathcal{S}(x_{\text{orig}}), y) + \lambda \cdot L_{\text{seg}}(\mathcal{S}(x_{\text{gen}}), y), \tag{6}$$

where $x_{\text{gen}}$ is a synthetic image sharing vascular structure with $x_{\text{orig}}$, and $\lambda > 0$ balances contributions from real and synthetic data. This consistency regularization improves robustness across diverse imaging conditions, enhancing segmentation performance on unseen datasets.

Additional implementation details, including hyperparameters and optimization strategies, are provided in the appendix.

## 5 Experimental Setup

We address data scarcity in retinal vessel segmentation by evaluating RLAD's ability to generate controllable, realistic fundus images and improve AV segmentation performance. Key evaluations include image realism (section 5.2), segmentation performance across backbones (section 5.3), SOTA comparisons (section 5.4), and ablation studies (section 6). We seek to address three key research questions:

- Can RLAD generate controllable, realistic retinal images?

- Does usage of RLAD-generated data enhance our AV segmentation model?

- How does our model perform compared to SOTA?

## 5.1 Evaluation Metrics

We evaluate the diffusion model's performance using the Fréchet Distance (FD), which compares the feature distributions of real and generated images. We compute it in the latent space of Inception-v3 (FID) [79] and RETFound [72] (RET-FD), a foundation model pre-trained on 1.6 million retinal images. RETFound likely offers a more accurate representation of retinal image-specific features, while Inception-v3 enables a comparison with previous work.

For AV segmentation, we use the Dice score to measure overlap between predicted and ground truth segmentations, averaged as $(\text{Dice}_A + \text{Dice}_V)/2$. This is complemented by the Intersection over Union (IoU) and centerline Dice (clDice) [80], which emphasizes vessel centerlines. Both Dice and clDice metrics are employed in RLAD ablation studies, with additional IoU and clDice results provided in the appendix. Notably, clDice offers a more nuanced evaluation by balancing sensitivity to both thin and large vessels.

## 5.2 Evaluation of Realism

We compare the FID scores achieved by RLAD with those of prior works (Table 3), using their publicly available models for image generation or reports their published results when the models were inaccessible. Notably, RLAD demonstrates superior performance by generating more realistic retinal fundus images, as evidenced by lower FID and RET-FD scores.

## 5.3 Integrating RLAD into Leading Backbones

In Table 2, we present the performance of RLAD-generated data on the AV segmentation task, evaluated using various backbones: DinoV2$_\text{small}$, RETFound, SwinV2$_\text{tiny}$, and SwinV2$_\text{large}$. The results are reported across Local, External, and OOD test sets. For comparison, the first rows include previously published state-of-the-art results under similar settings (i.e., Local, External, and OOD), where available.

RLAD consistently improves performance on External, and OOD test sets, demonstrating its backbone-agnostic advantages and its adaptability to in-domain and out-of-domain pretrained models. For example, integrating RLAD with RETFound yields performance improvements of 6.3%, 7.5%, and 6.8% on AV-WIDE, IOSTAR, and TREND, respectively. Notably, even when applied to the top-performing backbone, SwinV2$_\text{large}$, RLAD provides further performance gains of 0.2% on External and 0.4% in OOD datasets.

## 5.4 Segmentation performance vs SOTA

SwinV2$_\text{large}$, trained on our newly curated dataset and RLAD-generated data, surpasses previous state-of-the-art models across all Local, External, and OOD datasets, with the exception of RVD (Table 2). As illustrated in Figure 3, it demonstrates superior AV segmentation performance compared to SwinV2$_\text{large}$ trained solely on the UZLF dataset and LUNet, the best performing open-source

| Gen Model | Conditioning | FID↓ | RET-FD↓ |
|---|---|---|---|
| StyleGAN [81] | L | 138.0 | 120.8 |
| StyleGAN2 [82] | Demographics | 98.1 | 116.0 |
| StyleGAN2 [20][†] | AV | 122.8 | - |
| Pix2PixHD [20][†] | AV | 86.8 | - |
| RLAD (Our) | AV + L + CD | **30.3** | **79.7** |

Table 3: **Realism of Generated Images.** Lower FID and RET-FD on the DRTiD dataset indicate closer alignment with real data, reflecting realism. Notably, RLAD is able to generate controllable and more realistic retinal images. Models[†] trained and evaluated on private data.

model. Further quantitative and qualitative comparisons are included in the appendix. Moreover, a comprehensive analysis demonstrating the superiority of our model over previous state-of-the-art methods in estimating common vascular parameters is also provided in the appendix.

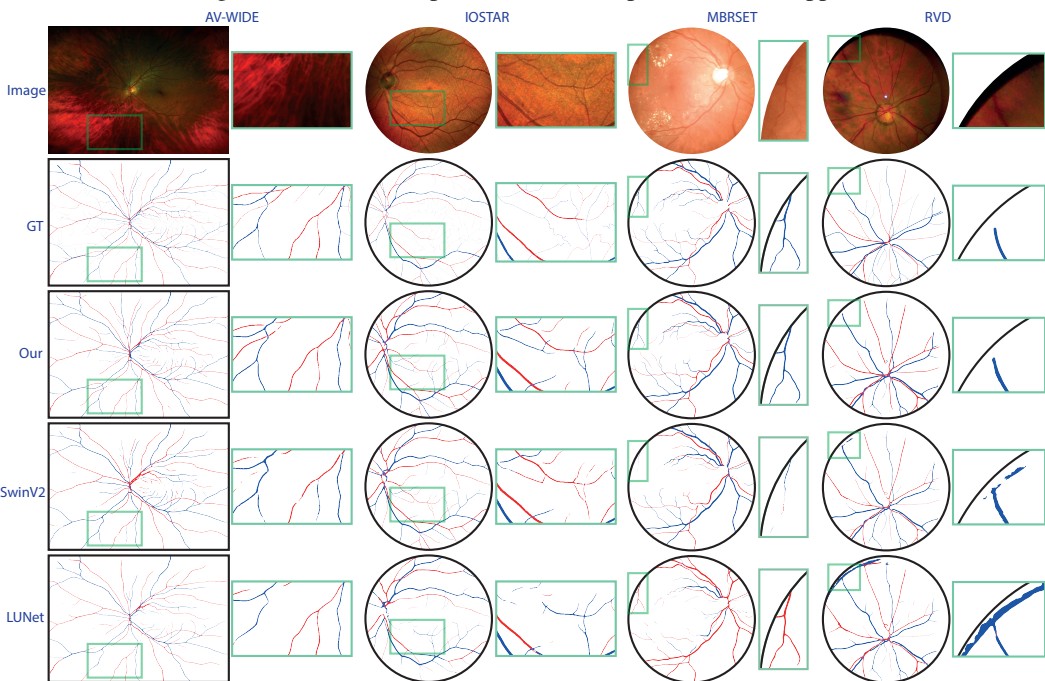

Figure 3: **Qualitative Example on the Segmentation Downstream Task.** Comparing our model's AV segmentation to a SwinV2$_{Large}$ [24] trained on the UZLF dataset and LUNet [9], a SOTA model, showcasing its superior performance across fundus images from various datasets.

## 6 Ablation studies

We analyze the effects of RLAD's components, training datasets, and pretraining objectives using SwinV2$_{tiny}$ as the baseline and Dice score unless stated otherwise.

**Training Datasets:** Starting with the UZLF dataset, we incrementally added our newly introduced datasets (Table 4). The Local test sets includes optic disc centered images, while External test sets mix optic disc and macula centered images. Adding macula-centered datasets GRAPE and MESSIDOR improved performance across Local, External and OOD test sets. Each dataset addition yielded incremental gains, with final improvements of +1.1%, +4.1%, and +8.3% for Local, External, and OOD, respectively.

**Pretraining Objective:** We evaluated how pretraining objectives (MAE, WCL, or both) influence our model's performance (see Table 5). Adding MAE or WCL individually improved the OOD Dice score from 68.9% to 69.2% and 69.4%, respectively, while combining them further increased clDice. These findings indicate that combining both strategies enhance model generalization.

**Conditioning on multiple layout components:** When learning a conditional distribution solely on AV, SwinV2$_{tiny+RLAD}$ achieved an average Dice score of 70.4% on the OOD datasets. In contrast, conditioning on multiple layout components (AV, CD, and L) improved performance to 70.8%. This highlights the advantage of leveraging a broader range of retinal fundus image features to enhance the learned distribution (see Table 5).

**Varying Generated Data Quantity:** We explored the impact of varying amounts of RLAD-generated samples: 0.5K (1 per real image), 1.5K (3 per real image), and 7.2K (15 per real image). Increasing generated samples improved the average OOD Dice (Table 6) and clDice (see appendix).

**Performance Gains of RLAD Relative to Dataset Size:** Figure 4 shows learning curves on OOD datasets for SwinV2$_{tiny}$ trained with and without RLAD synthetic data. Incorporating RLAD-generated data consistently improves performance across all datasets. For IOSTAR, RVD, DRIVE,

| Datasets | Size | Local | External | OOD |
|---|---|---|---|---|
| UZLF [53] | 184 | 82.1 | 75.5 | 60.6 |
| + GRAPE (Our†) | 81 | 82.6 | 78.1 | 65.2 |
| + MESSIDOR (Our†) | 67 | 82.8 | 78.9 | 66.6 |
| + ENRICH (Our*) | 111 | 83.1 | 79.2 | 67.0 |
| + MAGRABIA (Our†) | 69 | 83.1 | 79.2 | 67.2 |
| + PAPILA (Our†) | 78 | **83.1** | **79.6** | **68.9** |
| Δ | | +1.0 | +4.1 | +8.3 |

Table 4: **Impact of increasing the number of training datasets.** This table shows how adding newly introduced (*) or annotated (†) datasets to the SwinV2$_{tiny}$ training pipeline impact performance.

| PT | | FT | Local | | External | | OOD | |
|---|---|---|---|---|---|---|---|---|
| MAE | WCL | Gen | Dice | clDice | Dice | clDice | Dice | clDice |
| ✗ | ✗ | ✗ | 83.1 | 83.6 | 79.6 | 80.7 | 68.9 | 68.8 |
| ✓ | ✗ | ✗ | 83.1 | 83.6 | 79.6 | 80.8 | 69.4 | 69.2 |
| ✗ | ✓ | ✗ | 83.2 | 83.6 | 79.7 | 80.8 | 69.2 | 69.1 |
| ✓ | ✓ | ✗ | 83.2 | 83.6 | 79.6 | 80.8 | 69.4 | 69.3 |
| ✓ | ✓ | AV | 83.3 | 83.7 | 79.9 | 81.1 | 70.4 | 70.5 |
| ✓ | ✓ | AV + CD + L | **83.3** | **83.7** | **79.9** | **81.1** | **70.8** | **71.1** |
| Δ | | | +0.2 | +0.1 | +0.3 | +0.4 | +1.9 | +2.3 |

Table 5: **Pretraining Objective and Generation Method.** The top section shows baseline performance on our dataset, the middle highlights the impact of pretraining objectives, and the bottom examines AV conditioning versus AV + CD + L, with notable OOD improvements using AV + CD + L.

| # Gen | AV-WIDE | IOSTAR | DRIVE | RVD | TREND | MBRSET | OOD |
|---|---|---|---|---|---|---|---|
| 0.5K | 69.2 | 69.9 | 77.2 | 45.8 | 76.9 | 75.9 | 70.4 |
| 1.5K | 69.5 | 70.5 | 77.1 | 46.4 | 76.9 | 76.0 | 70.6 |
| 7.2K | 69.5 | 71.3 | 77.1 | 46.3 | 77.1 | 76.2 | **70.8** |

Table 6: **Quantity of Generated Data.** We evaluate the impact of increasing RLAD's generated data on performance, reporting Dice scores for each OOD dataset and their average performance.

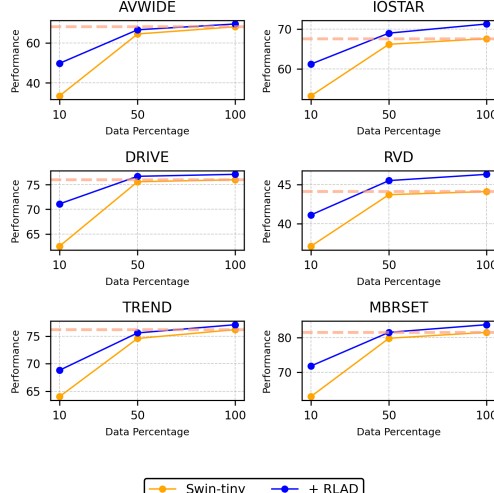

Figure 4: **RLAD Performance vs. Training Data Size.** The figure illustrates the learning curve of the SwinV2$_{tiny}$ [24] baseline on OOD datasets, demonstrating enhanced performance with RLAD-generated data. The data percentage reflects both real and generated samples, maintaining a 1:15 ratio (real:generated).

and MBRSET, the model trained with synthetic data outperformed the baseline while using less than 50% of the baseline's training data. The largest gains occurred in data-scarce scenarios, highlighting RLAD's effectiveness in enhancing performance.

# 7 Conclusion

This work presents RLAD, a novel diffusion-based framework designed to generate realistic and controllable retinal fundus images by conditioning on multiple layout components extracted from real-world data. Beyond image generation, RLAD proves to be a valuable tool for advancing downstream tasks. By incorporating the synthetic data generated by RLAD, we significantly enhance the training datasets for AV segmentation tasks, resulting in notable performance improvements across various visual backbones. This capability is particularly impactful in data-scarce scenarios, where access to comprehensive datasets is limited. Our findings highlight the potential of RLAD to drive innovation in medical imaging applications and improve segmentation outcomes. Future research could explore its application to other imaging modalities and investigate optimization strategies to further enhance its adaptability and scalability.

**Limitations and Societal Impact:** While RLAD improves generalization in retinal vessel segmentation, its effectiveness may be constrained by the quality of the generated images and the diversity of the training data. The approach may not fully generalize to imaging modalities or populations not represented in the training set. We demonstrated that the proposed framework may enhance clinical decision support for retinal image analysis, but care must be taken to avoid over-reliance on synthetic data and to monitor for biases that could affect underrepresented groups. Misapplication to non-target populations or imaging modalities could lead to incorrect diagnoses.

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
