601 # Appendix: Enhancing Retinal Vessel Segmentation Generalizationvia
602 **Layout-Aware Generative Modelling**

## Table of Contents

## A  Datasets

For our experiments, we utilized two distinct dataset combinations to support both the training and evaluation phases of our methodology.

The dataset tables provide a comprehensive summary of the key characteristics of each dataset, including the number of samples, the primary pathology—glaucoma (G), diabetic retinopathy (DR), age-related macular degeneration (AMD), or multiple different diseases (Multiple)—the imaging center, which is either disc (D) or macula (M), field of view (FOV), geographic region, and image resolution.

### A.1  Diffusion and Pretraining Datasets

The first combination involved non-annotated datasets used for training the RLAD model and pretraining segmentation models, as summarized in Table 7.

| Dataset | # Samples | Primary Pathology | Image Center | FOV (°) | Region | Resolution (px) |
|---|---|---|---|---|---|---|
| UZLF [53] | 184 | G | D | 30 | Belgium | 1444×1444 |
| GRAPE [47] | 81 | G | M | 50 | China | 1444×1444 |
| MESSIDOR [48] | 67 | DR | M | 45 | France | 1444×1444 |
| PAPILA [50] | 78 | G | D | 30 | Spain | 1444×1444 |
| MAGHREBIA [49] | 69 | – | M, D | 30 | Maghreb | 1444×1444 |
| ENRICH | 111 | G | D | 45 | Belgium | 1958×2196 |
| 1000images [54] | 973 | Multiple | D | 30 | China | 3000x3152 |
| DDR [55] | 12 519 | DR | M | 45 | China | 1728x2592 |
| EYEPACS [56] | 88 702 | DR | M | 45 | United States | VAR |
| G1020 [57] | 1020 | G | M | 45 | Germany | 2423x3004 |
| IDRID [58] | 516 | DR | M | 50 | India | 2848x4288 |
| ODIR [59] | 8000 | Multiple | M | 45 | China | 1296x1936 |

Table 7: **Summary of Datasets Used for Pretraining and RLAD Training.** This table lists the datasets used for pretraining segmentation models and training the RLAD framework. Key attributes include the number of samples, primary pathologies, imaging center type, field of view (FOV), geographic region, and resolution.

### A.2  Segmentation Datasets

The second combination comprised AV-annotated datasets, which were employed for training segmentation models on downstream tasks (Table 8) and for evaluating their performance (Table 9). Datasets annotated specifically for this study are marked with [†], while those introduced and annotated as part of this work are marked with [∗].

| Dataset | # Samples | Primary Pathology | Image Center | FOV (°) | Region | Resolution (px) |
|---|---|---|---|---|---|---|
| UZLF [53] | 184 | G | D | 30 | Belgium | 1444×1444 |
| GRAPE[†] [47] | 81 | G | M | 50 | China | 1444×1444 |
| MESSIDOR[†] [48] | 67 | DR | M | 45 | France | 1444×1444 |
| PAPILA[†] [50] | 78 | G | D | 30 | Spain | 1444×1444 |
| MAGHREBIA[†] [49] | 69 | – | M, D | 30 | Maghreb | 1444×1444 |
| ENRICH[∗] | 111 | G | D | 45 | Belgium | 1958×2196 |

Table 8: **Summary of Datasets Used for Downstream Segmentation Training.** This table lists the annotated datasets used for training segmentation models in downstream tasks. Attributes include the number of samples, primary pathologies, imaging center type, field of view (FOV), geographic region, and resolution. Datasets marked with [†] were annotated specifically for this work, and those marked with [∗] were both introduced and annotated here.

| | Dataset | # Samples | Primary Pathology | Image Center | FOV (°) | Region | Resolution (px) |
|---|---|---|---|---|---|---|---|
| **Local** | UZLF-test [53] | 56 | G | D | 30 | Belgium | 1444×1444 |
| | LES-AV [16] | 20 | G | D | 30 | Belgium | 1444×1444 |
| **External** | HRF [69] | 45 | DR, G | M | 45 | Germany | 2336×3504 |
| | INSPIRE [9, 62] | 15 | – | D | 30 | USA | 1444×1444 |
| | FIVES[†] [45] | 75 | DR, G, AMD | M | 45 | China | 1444×1444 |
| | UNAF [9, 63] | 15 | DR | D | 30 | Paraguay | 2056×2124 |
| **OOD** | AV-WIDE[†] [52] | 26 | – | D | Ultra wide | USA | 829×1531 |
| | IOSTAR [64] | 30 | – | M | 45 | Netherlands | 1024×1024 |
| | DRIVE [65, 66] | 40 | DR | M | 45 | Netherlands | 584×565 |
| | RVD [10] | 1270 | – | VAR | 30 | – | 1800x1800 |
| | TREND[†] [46] | 48 | H | M | 30 | Montenegro | 2560×2560 |
| | MBRSET[†] [51] | 30 | DR, G, AMD | M | 30 | Brazil | 1444×1444 |

Table 9: **Summary of Datasets Used for Segmentation Benchmark Evaluation.** This table categorizes datasets into in-domain (Local), near-domain (External), and out-of-domain (OOD) groups for evaluating segmentation performance. Attributes include the number of samples, primary pathologies, imaging center type, field of view (FOV), geographic region, and resolution. Datasets marked with [†] were annotated specifically for this work, and those marked with [*] were both introduced and annotated here.

## B    Training Hyperparameters

All experiments were conducted on 4 Nvidia A100 (40G) GPUs using bfloat16 precision. In each training the AdamW optimizer [83] and the Cosine Annealing scheduler [84] were uniformly applied. Beyond these constants, each training was characterized by its own distinct set of hyperparameters.

**RLAD Training:** comprised 84,000 training steps, with a learning rate $1e-4$ and and a batch size of 12.

**Segmentation Models Pretraining:** comprised 1 training epoch, with a learning rate $1.5e-4$ and and a batch size of 128.

**Segmentation Models Finetuning:** comprised 200 training epochs, with a learning rate $4e-4$. Other hyperparameters varied based on the backbone and are described in Table 10.

| Backbone | # Epochs | # Batch Size | Learning Rate | $\lambda$ |
|---|---|---|---|---|
| DinoV2$_{small}$ [71] | 200 | 12 | $4e-4$ | 1.0 |
| RETFound [72] | 200 | 12 | $4e-4$ | 0.1 |
| SwinV2$_{tiny}$ [73] | 200 | 12 | $4e-4$ | 0.1 |
| SwinV2$_{large}$ [73] | 200 | 2 | $4e-4$ | 0.1 |

Table 10: **Hyperparameters for the segmentation downstream task finetuning.**

## C    Additional Quantitative Results

In addition to the metrics reported in the main paper, we report Intersection over Union (IoU) and centerline Dice score (clDice) for SwinV2$_{Large + RLAD}$ versus the open -souce models. IoU measures the ratio of the intersection to the union of the predicted and ground truth segmentation masks, providing an additional evaluation of segmentation performance. The IoU is computed separately for arteries (A) and veins (V), and we report the average IoU across both classes $(\text{IoU}_A + \text{IoU}_V)/2$. This metric complements the Dice score by offering a stricter evaluation of overlap, particularly for challenging cases with smaller or less distinct structures. Table 13 shows that our model outperform all open-source baseline for both clDice and IoU across all datasets, except the DRIVE where VascX [70] get higher IoU performance.

| Backbone | External | | | | | | | | OOD | | | | | | | | | | | |
|---|---|---|---|---|---|---|---|---|---|---|---|---|---|---|---|---|---|---|---|---|
| | HRF | | INSPIRE | | FIVES | | UNAF | | AV-WIDE | | IOSTAR | | DRIVE | | RVD | | TREND | | MBRSET | |
| | clDice | IoU | clDice | IoU | clDice | IoU | clDice | IoU | clDice | IoU | clDice | IoU | clDice | IoU | clDice | IoU | clDice | IoU | clDice | IoU |
| Little W-Net [7] | 53.3 | 41.5 | 70.7 | 55.6 | 71.9 | 59.0 | 68.5 | 52.5 | 41.1 | 28.1 | 26.6 | 19.3 | 59.7 | 44.4 | 32.1 | 22.2 | 51.9 | 36.9 | 35.2 | 34.6 |
| Automorph [34] | 76.7† | 63.3† | 71.5 | 55.3 | 72.1 | 57.9 | 66.3 | 49.9 | 49.9 | 33.9 | 52.3 | 38.4 | 77.3† | 64.1† | 31.6 | 22.6 | 65.3 | 50.4 | 62.0 | 47.8 |
| VascX [70] | 73.1 | 61.0 | 75.3 | 60.0 | 79.1 | 67.6 | 74.3 | 57.9 | 49.7 | 34.1 | 49.0 | 35.6 | 75.9 | 63.5 | 39.7 | 28.1 | 69.6 | 56.4 | 73.4 | 58.3 |
| LUNet [9] | 72.8 | 58.1 | 76.4 | 64.9 | 82.6 | 75.9 | 76.7 | 59.5 | 65.5 | 53.4 | 52.1 | 40.2 | 71.3 | 55.4 | 36.1 | 22.4 | 69.6 | 55.9 | 64.0 | 48.0 |
| SwinV2$_{Large + RLAD}$ (Our) | 81.1 | 67.5 | 83.0 | 65.5 | 86.9 | 77.7 | 78.3 | 61.4 | 73.2 | 55.7 | 73.0 | 59.8 | 80.3 | 62.9 | 49.1 | 33.0 | 77.9 | 63.8 | 86.8 | 76.0 |

Table 13: **Additional RLAD Results**. Quantitative comparison of SwinV2$_{Large + RLAD}$ versus open source models. Performance is the average clDice/IoU for artery and vein. † indicates data leakage during training.

# D  Additional Ablation Results

Additional ablation results on the impact of the scale of the generated samples using clDice score are shown in Table 11. It shows that using more RLAD-generated samples also increased the average OOD performance for the clDice score.

| # Gen | AV-WIDE | IOSTAR | DRIVE | RVD | TREND | MBRSET | OOD |
|---|---|---|---|---|---|---|---|
| 0.5K | 70.9 | 67.5 | 79.6 | 46.2 | 75.9 | 83.7 | 70.6 |
| 1.5K | 70.9 | 68.2 | 79.6 | 46.8 | 76.0 | 84.0 | 70.9 |
| 7.2K | 70.9 | 69.0 | 79.7 | 46.6 | 76.2 | 84.2 | **71.1** |

Table 11: **Quantity of Generated Data.** We evaluate the impact of increasing RLAD's generated data on performance, reporting clDice scores for each OOD dataset and their average performance.

# E  Impact of the Layout Extractor

RLAD is trained on an approximation of the layout extracted by a deep learning model, rather than relying on a ground truth conditioning. This enables RLAD to learn a distribution $p_\theta(x_{t-1}|\widetilde{\text{layout}}, x_t)$ instead of $p_\theta(x_{t-1}|\text{layout}, x_t)$, allowing the model to adapt to noisy conditioning. Consequently, RLAD exhibits a degree of robustness to the errors typically made by the layout extractor. Figure 6 illustrates this with intentionally corrupted images, generated by applying a random masking strategy. While the extracted blood vessels are impacted by the corruption, the final images generated by RLAD remain relatively unaffected, provided the density of the masks is limited. This robustness aligns with the known limitations of current retinal blood vessel segmentation models. Thus, we assume that the performance of the Layout Extractor remains a relatively unimportant factor (for small performance differences), given that its limitations will be mitigated by the diffusion model.

# F  Additional Qualitative Results

In Figure 5, we display some additional qualitative examples of our model compared to a SwinV2large baseline and a SOTA open-source model LUNet. We can see that our model more accurately segment the blood vessels of the DRIVE and TREND datasets.

# G  Vascular Parameters Estimation

Vascular parameters were estimated using the PVBM toolbox [17], including area (Area), tortuosity indices (TI, TOR), length (LEN), branching angles (BA), key vascular points (SPoints, EPoints, BPoints), fractal dimensions (D0, D1, D2, SL), and retinal metrics (CRAE/CRVE, AVR). Parameters were evaluated on OOD datasets by computing Pearson correlations between ground-truth and estimated values, with final scores representing averages across datasets and vascular structures (arteries/veins).

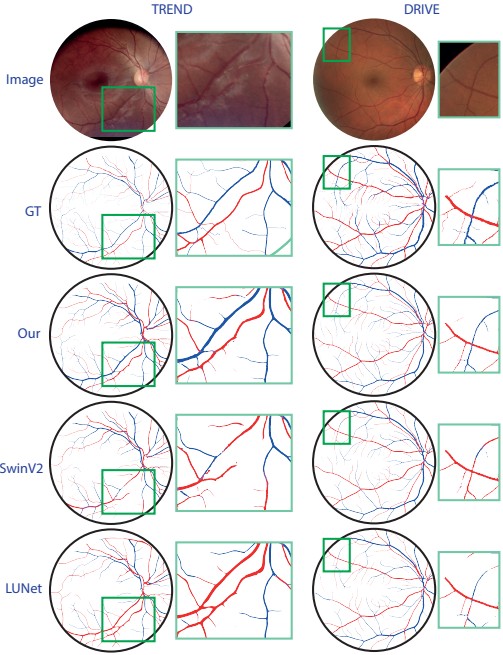

Figure 5: **Qualitative Example on the Segmentation Downstream Task.** Comparing our model's AV segmentation to a SwinV2$_{large}$ [73] trained on the UZLF dataset and SOTA model LUNet [9], showcasing its superior performance across diverse fundus images.

| Vascular Parameters | Little W-Net | Automorph | VascX | LUNet | Our |
|---|---|---|---|---|---|
| Area | 55.7 | **73.2** | 69.9 | 61.3 | 71.4 |
| TI | 46.3 | 61.3 | 62.9 | 59.3 | **71.7** |
| TOR | 45.6 | 53.9 | 61.7 | 60.6 | **68.8** |
| LEN | 56.8 | 69.3 | 68.9 | 68.5 | **75.5** |
| BA | 24.3 | 45.5 | 44.8 | 38.6 | **51.5** |
| SPoints | 41.6 | 56.4 | 56.7 | 55.4 | **62.2** |
| EPoints | 53.7 | 70.1 | 71.3 | 68.3 | **77.7** |
| BPoints | 39.4 | 55.3 | 55.8 | 53.1 | **65.2** |
| D0 | 56.0 | 59.8 | 65.3 | 61.6 | **69.0** |
| D1 | 60.9 | 68.3 | 73.2 | 72.7 | **80.7** |
| D2 | 48.7 | 54.1 | 58.7 | 60.8 | **70.0** |
| SL | 48.8 | 53.8 | 54.3 | 59.0 | **63.6** |
| CRE$_H$ | 55.1 | 66.0 | 66.8 | 69.9 | **75.8** |
| CRE$_K$ | 52.1 | 65.5 | 62.1 | 67.4 | **75.0** |
| AVR$_H$ | 66.7 | 74.3 | 78.9 | 78.2 | **81.0** |
| AVR$_K$ | 31.4 | 41.9 | 44.1 | 47.4 | **52.9** |
| Average | 48.9 | 60.5 | 61.4 | 62.2 | **69.5** |

Table 12: **RLAD Vascular Parameters Results**. Quantitative comparison of SwinV2$_{Large + RLAD}$ (Our) versus open-source models. Performance is reported as the average Pearson correlation coefficient in estimating vascular parameters across OOD datasets.

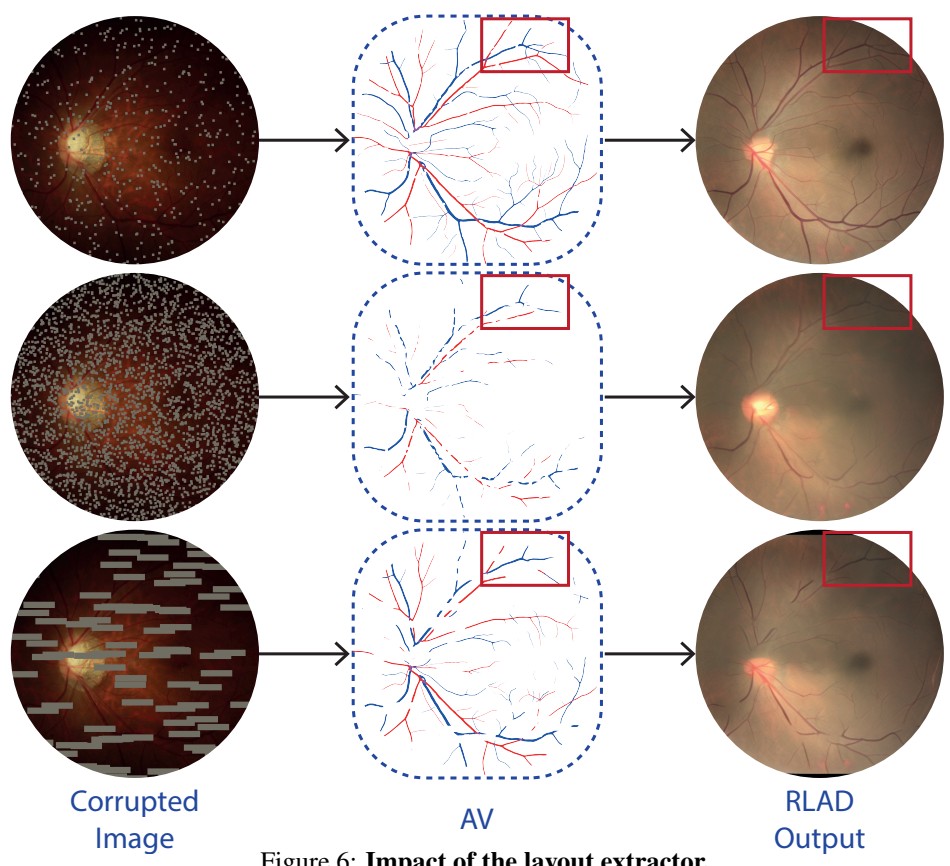

Corrupted Image    AV    RLAD Output

Figure 6: **Impact of the layout extractor.**