# OpenReview forum: "Enhancing Retinal Vessel Segmentation Generalization via Layout-Aware Generative Modelling"
_NeurIPS.cc/2025/Datasets_and_Benchmarks_Track — Submitted to NeurIPS 2025 Datasets and Benchmarks Track_

### Official Review · Reviewer_hNs4 · 2025-06-22

**Rating:** 4
**Confidence:** 4

**Summary:**

This paper introduces RLAD (Retinal Layout-Aware Diffusion), a diffusion-based framework that conditions retinal fundus image synthesis on multiple anatomical layouts (artery/vein, optic disc/cup, lesions) to generate paired image–segmentation data. The authors also assemble REYIA, a new multi-source dataset of 585 manually annotated retinal images. Experiments show that integrating RLAD-generated images into segmentation training yields up to an 8.1% improvement under distribution shift.

**Dataset Code Accessibility:**

Partly

**Dataset Code Comments:**

While the Kaggle URL is publicly reachable, three files are inaccessible per the compliance report, and no direct URLs exist for the PNG/JPEG assets.

**Ethical Comments:**

- It is unclear whether patients whose images contributed to public sources have consented to use—and synthetic derivative generation—of their data.

- Without demographic metadata on annotators or subjects, synthetic images may perpetuate unseen biases across populations.

- Also, without RAI metadata fields like rai:dataBiases and rai:dataSocialImpact, ethical auditing is hindered.

**Ethical Considerations:**

Yes, there are ethics concerns that require attention by the authors

**Ethics Flags:**

["Data privacy, copyright, and consent", "Data quality and representativeness", "Discrimination, bias, and fairness"]

**Final Justification:**

I appreciate the authors’ efforts in addressing my remaining concerns on Inter-Annotator Agreement, and I have adjusted my rating.

**Limitations Weaknesses:**

- The autogenerated compliance report flags “Accessible Data Files ⚠️ Inaccessible: 3 files”, indicating broken or missing URLs for some image assets on Kaggle.
- There are also missing responsible-AI metadata: The Croissant JSON lacks rai: fields (e.g. dataCollection, annotatorDemographics), reducing transparency around consent, annotation protocols, and bias assessment.
- While the ENRICH subset had IRB approval, the paper does not describe expert review or inter-annotator agreement for the nine sources’ vessel masks, leaving the risk of inconsistent segmentations.
- Both the methodology and its dataset are restricted to fundus imaging, and their effectiveness on other ocular or vascular imaging modalities has yet to be explored.

**Strengths Contributions:**

+ The paper is well-organized, figures and tables are informative, and the code is publicly accessible, facilitating reproducibility.
+ Across three backbones (DinoV2, RETFound, SwinV2) and in-domain, external, and OOD test sets, RLAD shows a boost in segmentation performance (Table 2) .
+ The creation and open release of a 585-image, AV-segmented retinal fundus dataset drawn from nine sources fills a gap in publicly available multi-center AV data.
+ The authors also ablate the impact of pretraining objectives (MAE, WCL), conditioning components, dataset additions, and generated-data volume.

---

> ### Author Rebuttal · Authors · 2025-07-29
>
> Thank you for your detailed review and helpful observations. We appreciate your recognition of the clear presentation, strong performance across architectures and test sets, and the value of releasing a multi-source annotated dataset. Your comments on data accessibility, metadata, and ethical considerations were especially valuable and have been carefully addressed in the revised paper. Our responses are presented below in Q&A format.
>
> ## 1. Data Accessibility Issues
>
> We acknowledge the reviewer's concern regarding data accessibility and will reupload the datasets entirely to Kaggle to ensure proper access for the research community.
>
> ## 2. Responsible AI Metadata
>
> We acknowledge the reviewer's request for comprehensive Responsible AI documentation and we will update the Croissant JSON to include detailed RAI fields covering data collection procedures, annotator demographics, potential biases, and social impact assessment as shown in the provided JSON structure.
>
> ```json
> {
>  "rai:dataCollection": {
>    "consentProcedure": "IRB approval with consent waiver",
>    "dataProtection": "HIPAA/GDPR compliant anonymization",
>    "collectionMethod": "Clinical imaging during routine care"
>  },
>  "rai:annotatorDemographics": {
>    "expertise": "10 ophthalmology students experienced in microvascular research and a senior annotator (medical doctor, resident in ophthalmology and graduated from a PhD with focus on retinal biomarkers)",
>  },
>  "rai:dataBiases": {
>    "geographic": "15 countries, potential underrepresentation of African populations",
>    "pathology": "DR, glaucoma, AMD, healthy",
>    "imaging": "Multiple fundus camera types"
>  },
>  "rai:dataSocialImpact": {
>    "benefits": "Improved model's blood vessel segmentation to a variety of population samples",
>    "risks": "The data may not fully represent imaging modalities or populations not represented within our dataset."
>  }
> }
> ```
>
> ## 3. Expert Review and Inter-annotator Agreement
> We acknowledge the reviewer's concern regarding annotation consistency across the nine data sources and will add the following clarification to the paper:
> “REYIA ensures annotation quality through a rigorous two-round quality control protocol: initial vessel segmentation by ten junior annotators (ophthalmology students with microvascular research experience) using Lirot.ai software [1], corrected and validated by a senior medical doctor (ophthalmology resident with PhD specialization in retinal vasculature analysis). Arteriole-venule discrimination was based on established morphological criteria including vessel darkness (venules appearing darker), central light reflex visibility (more prominent in arterioles), vessel caliber differences (venules typically thicker), anatomical positioning patterns near the optic disc (alternating arrangement), and crossing probability rules (same vessel types rarely cross). This systematic quality control and annotation approach, combined with its unprecedented diversity, ensures high inter-annotator consistency and makes REYIA the most representative and reliable dataset for real-world AV segmentation model development. All but the ENRICH fundus image datasets are open access. For the ENRICH dataset, ethics approval was obtained from UZ Leuven Ethics Committee (S60649) with informed consent waiver due to retrospective nature and complete data anonymization.”
>
> ## 4. Modality Limitations
> We acknowledge the reviewer's concern regarding modality limitations and will add the following clarification to the paper: our methodology is focused on  color fundus photography due to layout extractors being trained exclusively on fundus data and the substantial annotation time requirement of approximately 45 minutes per image for complete AV segmentation. Extension to other modalities such as OCT or fluorescein angiography would require retraining all components and is beyond the scope of our current research work.
>
>
>
> ## 5. Ethical Consent & Demographic Metadata:
> A detailed ethical consent will be included in the kaggle link of the dataset, containing the link to original papers with dedicated ethical consent approval.
> “All fundus image datasets used in this study were open access, except for the ENRICH subset. For the ENRICH data, ethical approval was obtained from the UZ Leuven Ethics Committee (S60649), with a waiver of informed consent due to the retrospective nature of the study and complete data anonymization. Similarly, training of RLAD was performed using a total of X open-access datasets, as listed in Table Y. Consequently, all data used in our experiments were either openly available in accordance with the respective datasets' ethical approvals or waivers, or, in the case of ENRICH, were used and is made open access under a new formal ethical waiver procedure.”
> We acknowledge the lack of demographic data as a limitation and will add a discussion on it in the limitation part of the camera ready version.
>
> ---
>
> Thanks again for your thoughtful review. If your concerns have been addressed, we would be grateful for your reconsideration of the score. We’re happy to clarify any remaining issues.

---

> > ### Comment · Reviewer_hNs4 · 2025-08-03
> >
> > One remaining concern is about inter-annotator agreement; it would also strengthen your paper to report a quantitative inter-annotator agreement metric—such as Cohen’s κ—to substantiate the consistency of your two-round quality-control process.

---

> > > ### Author Response · Authors · 2025-08-04
> > >
> > > ### Inter-Annotator Agreement
> > >
> > > To address the concern regarding annotation consistency, we report **inter-annotator agreement** using **Cohen's κ (kappa)**. This was computed across all datasets for which we performed our two-round quality-control process. The results demonstrate **strong consistency** between annotators.
> > >
> > > #### Per-Dataset Cohen's κ Scores
> > >
> > > | Dataset   | Artery κ | Vein κ  |
> > > |-----------|----------|---------|
> > > | AVWIDE    | 0.9474   | 0.9467  |
> > > | ENRICH    | 0.9981   | 0.9981  |
> > > | FIVES     | 0.9898   | 0.9929  |
> > > | GRAPE     | 0.9985   | 0.9982  |
> > > | MAGREB    | 0.9925   | 0.9911  |
> > > | MBRSET    | —        | —       |
> > > | MESSIDOR  | 0.9918   | 0.9935  |
> > > | PAPILA    | 0.9841   | 0.9848  |
> > > | TREND     | 0.9684   | 0.9676  |
> > >
> > > > **Note:** For the MBRSET dataset, junior annotations were not saved. As a result, we were unable to compute inter-annotator agreement for this dataset.
> > >
> > > #### Global Summary
> > >
> > > | Metric     | Artery κ | Vein κ |
> > > |------------|----------|--------|
> > > | **Mean**   | 0.9893   | 0.9898 |
> > > | **Std Dev**| 0.0221   | 0.0223 |
> > > | **Min**    | 0.7816   | 0.8142 |
> > > | **Max**    | 1.0000   | 1.0000 |
> > >
> > > These results confirm a **high degree of annotation reliability**, supporting the robustness of our two-stage annotation and validation pipeline.

---

> > > > ### Comment · Reviewer_hNs4 · 2025-08-04
> > > >
> > > > I thank the authors for addressing my remaining concerns and have updated my rating accordingly.

---

> > > > > ### Comment · Reviewer_hNs4 · 2025-08-05
> > > > > **Mandatory Acknowledgement by Reviewer hNs4**
> > > > >
> > > > > I have read the author rebuttal and considered all raised points., I have engaged in discussions and responded to authors., I have filled in the "Final Justification" text box and updated "Rating" accordingly (before Aug 13) that will become visible to authors once decisions are released., I understand that Area Chairs will be able to flag up Insufficient Reviews during the Reviewer-AC Discussions and shortly after to catch any irresponsible, insufficient or problematic behavior. Area Chairs will also be able to flag up during Metareview grossly irresponsible reviewers (including but not limited to possibly LLM-generated reviews)., I understand my Review and my conduct are subject to Responsible Reviewing initiative, including the desk rejection of my co-authored papers for grossly irresponsible behaviors. https://blog.neurips.cc/2025/05/02/responsible-reviewing-initiative-for-neurips-2025/

---

### Official Review · Reviewer_25Zx · 2025-06-28

**Rating:** 4
**Confidence:** 5

**Summary:**

This paper introduces Retinal Layout-Aware Diffusion (RLAD), a controllable diffusion-based framework that generates synthetic retinal images conditioned on key structural components, such as blood vessels. By preserving vascular structure while varying features like lesions and the optic disc, RLAD creates diverse, layout-aware image–segmentation pairs. Applied to retinal fundus imaging, RLAD improves vessel segmentation performance by up to 8.1%. They have releases REYIA, a new dataset of 585 manually segmented retinal images. All code and data will be publicly available to support reproducibility and further research.

**Additional Feedback:**

The author names and affiliations were included in the submitted manuscript, which is not in full compliance with the double-blind review policy.

**Dataset Code Accessibility:**

Yes

**Dataset Code Comments:**

I have checked the datasets and all the images with labels are available.

**Ethical Comments:**

There are no significant ethical concerns, as all datasets included are publicly available and have been released with appropriate approvals and consent for research use.

**Ethical Considerations:**

No, there are no or only very minor ethics concerns

**Final Justification:**

I have reviewed the authors rebuttals and other reviewers' comments, prefer to keep the same rating. The minor thing is that the lesions they have mentioned are related to diabetic retinopathy. It would be more useful for it instead of other retinal diseases.

**Limitations Weaknesses:**

Only the segmentation annotations are shared publicly, while the other imaging data used for pretraining and synthesis remain unavailable (not directly available). The manuscript provides no details on the types of lesions included or the methods used to extract or represent the optic disc information from the images.

**Strengths Contributions:**

It introduces Retinal Layout-Aware Diffusion (RLAD), a novel generative framework capable of producing diverse yet anatomically accurate retinal images. By conditioning image generation on multiple key structural components, such as blood vessels, RLAD ensures the preservation of critical semantic information while enabling variability in less constrained regions. This fine-grained control supports more realistic and meaningful data augmentation.

The study demonstrates that training with RLAD-generated images leads to consistent performance improvements in retinal vessel segmentation across a range of state-of-the-art models, underscoring the method’s generalizability and practical impact.

The authors contribute to the community by releasing REYIA, a new dataset of 585 carefully annotated retinal images, providing a valuable benchmark for future research.

---

> ### Author Rebuttal · Authors · 2025-07-29
>
> Thank you for your encouraging and constructive review. We appreciate your recognition of the method's ability to generate diverse yet structurally accurate retinal images, its impact on segmentation across models, and the contribution of the REYIA dataset as a benchmark. Your comments on data availability were helpful in improving clarity and accessibility. All concerns have been addressed in the revised version; responses follow in Q&A format.
>
> ## 1. Data Availability
>
> We acknowledge the reviewer's inquiry regarding data availability.
>
> **Newly Annotated Data (REYIA):** All 585 images and segmentations are available on Kaggle, including metadata such as resolution, FOV, and pathology labels.
>
> **Generated Synthetic Data:** The 7,230 generated images will be added to the Kaggle repository.
>
> **Public Datasets Used:** These are referenced in the supplementary material and are open access.
>
> ## 2. Lesion Types
>
> We acknowledge the reviewer's request for clarification regarding the specific lesion types included in our dataset and will add the following information to the paper: "The lesion types include the lesion encompass soft exudates, hard exudates, hemorrhages, and microaneurysms."
>
> ## 3. Layout Extraction Models
>
> We acknowledge the reviewer's comment regarding the methods used to extract optic disc and lesion information. As mentioned in the paper, we used the disc segmentation method from Fhima et al. [1,2] and the lesion segmentation approach developed by Men et al. [3]. Both model are openly available on the Python Vasculature Biomarker toolbox (PVBM) Github. We will add a link to the models in the camera ready version.
>
> [1] Jonathan Fhima, Jan Van Eijgen, Anat Reiner-Benaim, Lennert Beeckmans, Or Abramovich, Ingeborg
> Stalmans, and Joachim A Behar. Computerized analysis of the eye vasculature in a mass dataset of digital
> fundus images: the example of age, sex and primary open-angle glaucoma, Ophthalmology Science, 5(5):100778, 2025.
>
> [2] Or Abramovich, Hadas Pizem, Jonathan Fhima, Eran Berkowitz, Ben Gofrit, Meishar Meisel, Meital Baskin, Jan Van Eijgen, Ingeborg Stalmans, Eytan Z. Blumenthal and Joachim A. Behar. GONet: A Generalizable Deep Learning Model for
> Glaucoma Detection, IEEE Transactions on Biomedical Engineering, pp. 1–11, 2025.
>
> [3] Yevgeniy Men, Jonathan Fhima, Leo Anthony Celi, Lucas Zago Ribeiro, Luis Filipe Nakayama, and Joachim A. Behar. Deep learning generalization for diabetic retinopathy staging from fundus images. Physiological Measurement, 13(1), 2025.
>
>
>
> ## 4. Author names and affiliation
>
> We acknowledge the reviewer's concern regarding anonymity. The datasets and benchmark track is a single-blind submission according to NeurIPS 2025 guidelines (https://neurips.cc/Conferences/2025/CallForDatasetsBenchmarks), meaning author identities are known to reviewers but not vice versa. However, we have made efforts to maintain anonymity where possible throughout our submission to follow standard academic practices.
>
> ---
>
> Thank you again for your valuable feedback. If our clarifications resolve your concerns, we would be grateful if you would consider revising your evaluation. We're happy to provide further details if needed.

---

### Official Review · Reviewer_adNS · 2025-07-03

**Rating:** 4
**Confidence:** 5

**Summary:**

In medical imaging, particularly retinal vessel segmentation, data scarcity and variability in imaging conditions remain persistent limitations. However, exist generative models encounter challenges in preserving anatomical fidelity and issues with training stability. To address these limitations, authors introduce a novel retinal-layout-aware generative model (RLAD) that synthesizes diverse retinal images while preserving some key retinal layout components, like optic disc.  Through the experiments, demonstrating consistent segmentation performance improvements across state-of-the-art 52 architectures using RLAD-generated data. Finally, introducing REYIA, a very big multi-source collection of datasets for AV-segmented retinal fundus images.

**Dataset Code Accessibility:**

Partly

**Dataset Code Comments:**

The dataset is in a well-organized manner. But a detailed instruction file is missing.

**Ethical Considerations:**

No, there are no or only very minor ethics concerns

**Final Justification:**

I appreciate the authors' effort to address my concerns. However, I still hold the opinion that the lack of comparison with the CNN-based methods damaged the persuasiveness of the experimental results. So I keep my previous rating unchanged.

**Limitations Weaknesses:**

1. Line 67-71, Not clearly illustrate the limitation and reasons of Go's work.
2. Figure 4, What does the pink dotted line in Figure 4 represent?
3. The method lacks innovation and merely utilizes the existing pre-trained models to perform data augmentation.
4. During the generation process, for the distributions in the distribution perception, the author used three types: optic cup and disc, arteries and veins, and lesions. However, no discussion was made on the significance differences of these three distributions in the final result generation. This is because the proportions of these three distributions vary throughout the entire retina.
5. In medical images, especially in ophthalmic images, when studying the correlation between different types of blood vessels (such as arteries and veins) and diseases, the morphological changes of the vessels are of great significance. How do the authors ensure the authenticity and consistency of the generated blood vessels during the generation process?
6. In Figure 2, in the second row of the second subgraph, the types of blood vessels before and after are inconsistent. How did the author impose constraints in the model?
7. When encoding into the latent representation, how did the authors handle the two different types of vessel labels (arterial and venous), the two different types of labels (optic cup and optic disc), and the different lesion labels?
8. Is there any verification on the mature methods such as CNN, regarding their performance when trained using manually annotated images and when incorporating generated datasets for training on the same test set? Will the generated datasets introduce noise during training, thereby causing a decline in performance?

**Strengths Contributions:**

Introducing REYIA, the largest multi-source collection of datasets for AV-segmented retinal fundus images. The dataset organized by the author can significantly promote the development of datasets in the field of arteriovenous segmentation, especially since the existing datasets of arteriovenous data are relatively small. The author used the latest Diffusion Model to generate diverse fundus images based on the existing annotations of arteriovenous vessels, enriching the ophthalmic dataset. The author utilized three key layout components as guidance, which enabled the generation of more realistic images. The author conducted sufficient comparative experiments and ablation experiments to verify that the generated dataset is closer to the real dataset than the SOTA methods, and the generated dataset can enhance the performance of existing models.

---

> ### Author Rebuttal · Authors · 2025-07-29
>
> Thank you for your detailed review and constructive comments. We're glad you found value in the use of our Retinal Layout-Aware Diffusion model, the incorporation of multiple structural components, and the scale and organization of the REYIA dataset. Your suggestions helped us clarify key aspects of our approach and better explain our choices. We have addressed all the concerns you pointed out and will integrate them in the revised camera ready version.
>
> ## 1. Limitations of Go's Work (Lines 67-71)
>
> We acknowledge the reviewer's request for clearer positioning relative to Go et al.'s work. Go et al. [2024] proposed a two-stage approach combining diffusion models for AV mask generation with conditional GANs for image synthesis, with limitations including performance parity without improvement over real data baselines, evaluation limited to 2 datasets (one proprietary), insufficient diversity in generated AV masks propagating to synthesized images, and no demonstration of improved downstream task performance. Our RLAD approach addresses these through direct conditioning on real vessel structures and demonstrated performance gains across 12 diverse test sets.
>
> ## 2. Figure 4 Pink Dotted Line
>
> We acknowledge the reviewer's request for clarification regarding the pink dotted line in Figure 4. The pink dotted line represents the performance asymptote achieved using only real data, enabling clear visualization of the performance ceiling without synthetic augmentation and highlighting RLAD's contribution in breaking through this limit. We will add the following clarification to the caption: "The pink dotted line represents the performance asymptote achieved using only real data for model training."
>
> ## 3. Method Innovation
>
> We acknowledge the reviewer’s request for clarification regarding RLAD’s technical contributions. While RLAD is implemented on top of the DiT backbone, it adds the following novel components:
> -  Multi-layout conditioning: simultaneous, independent embeddings for artery/vein (AV), optic disc/cup (CD) and lesion (L) masks, allowing the generator to respect multiple anatomical constraints at once.
> -  UI–token interface: a per-component “present / absent” token that enables classifier-free guidance on any subset of the three layouts during both training and sampling.
> -  Paired-data synthesis loop: the same AV mask fed to RLAD is reused as a perfect ground-truth label, producing large volumes of image–label pairs without extra annotation.
>
> Thanks to these additions—build on top of vanilla DiT—we were able to:
> -  Create controllable, anatomically faithful fundus images in which vessels stay fixed while disc or lesions are freely varied.
> -  Generate 7 230 high-quality, label-aligned samples from only 480 re-labeled reals, boosting out-of-domain AV-segmentation Dice by up to 8.1 pp.
>
> ## 4. Distribution Perception Significance
>
> RLAD processes each of the three key retinal layout components—artery/vein (AV), optic cup/disc (CD), and lesions (L)—as independent conditioning inputs. During training, each component is embedded via its own projection head, and is either provided as extracted from a real fundus image or, with a certain probability, masked out and replaced by a neutral ("black") embedding; this masking is indicated by corresponding UI tokens. At sampling, RLAD's classifier-free guidance uses these UI tokens to selectively enforce or relax anatomical constraints: when a given component (e.g., AV, CD, or L) is included, the generated image will closely adhere to that structure, while omitted components are plausibly synthesized or left unconstrained. This design allows AV conditioning to anchor vascular topology, CD maps to dictate global disc/cup placement, and lesion masks to control the presence or absence of pathology—all within a single unified framework, enabling fine-grained, layout-aware control over generated retinal images.
>
> ## 5. Vessel Authenticity and Consistency
>
> We acknowledge the reviewer's remark regarding vessel authenticity and consistency in our generated images. Our validation approaches confirm vessel authenticity through quantitative metrics: Dice score improvements of up to +8.1% (IOSTAR), +3.7% (TREND), and +2.2% (RVD), along with centerline Dice (clDice) improvements that confirm topology preservation in the generated vessel structures.
>
> ## 6. Figure 2 Vessel Type Inconsistency
>
> We thank the reviewer for highlighting this. Indeed there was some misalignment that had been introduced during the creation of the figure. We will correct the figure.
>
> ## 7. Latent Representation Encoding
>
> We acknowledge the reviewer's request for clarification regarding the latent representation encoding process. The VAE processes 3 multi-channel inputs where each label is encoded in different colors to preserve semantic information:
>
> - **AV component** uses red for artery masks and blue for vein masks
> - **LESION component** encodes soft/hard exudates as pink/orange, hemorrhages as white, and microaneurysms as cyan
> - **CD component** uses yellow for cup masks and green for disc masks.
>
> ## 8. CNN Verification
>
> In this work we focused on transformer architectures as they are the state-of-the-art architectures for AV segmentation.
>
> ## 9. Dataset instruction
>
> We acknowledge the reviewer's feedback regarding the need for detailed dataset instructions and we will add an example notebook to the Kaggle database demonstrating how to load the real images, generated images, and their corresponding segmentations.
>
> ---
>
> We appreciate your engagement with our work. If the revisions meet your expectations, we would appreciate your consideration of a higher rating. Please don't hesitate to reach out for further clarification.

---

### Official Review · Reviewer_UU79 · 2025-07-21

**Rating:** 4
**Confidence:** 5

**Summary:**

The paper identifies the lack of annotated data as a key limitation for generalization in retinal image segmentation, and addresses this by using a generative model to synthesize new image-segmentation pairs. The authors also present REYIA, a new multi-source dataset with 586 manually segmented images, curated from nine public datasets and a newly collected one (ENRICH). Experimental results demonstrate that training with synthetic data generated by RLAD significantly improves segmentation performance across multiple models and cross-domain test sets.

**Additional Feedback:**

The paper introduces a method and dataset, it would benefit from clearer positioning—whether the main contribution is methodological (RLAD) or dataset-related (REYIA). The dataset construction process and annotation details should be more thoroughly described.

**Dataset Code Accessibility:**

Partly

**Dataset Code Comments:**

The REYIA dataset is available on Kaggle. However, the code is not hosted on a public GitHub repository. Instead, it is provided as a large 10GB download via Google Drive, which makes it difficult for others to conveniently browse the code. Hosting the code on a public platform like GitHub would improve accessibility and reproducibility.

**Ethical Comments:**

The methods and datasets used in this work do not raise any apparent ethical, societal, or privacy concerns.

**Ethical Considerations:**

No, there are no or only very minor ethics concerns

**Final Justification:**

All my issues have been addressed. Now I clearly understand the data contributions made by the article.

**Limitations Weaknesses:**

1. Misalignment Between Submission Track and Core Contribution: While the paper is submitted under the dataset track, its primary contribution appears to be the development of a generative method—Retinal Layout-Aware Diffusion (RLAD)—for generating layout-conditioned retinal images. The experimental section focuses extensively on demonstrating the impact of RLAD-generated data on improving segmentation performance. The Related Work section mainly discusses generative and segmentation models, with no comprehensive review of existing retinal AV segmentation datasets or prior works in synthetic data augmentation for medical imaging.
2. If the REYIA dataset is the reason for submission to the dataset track, the paper lacks a detailed evaluation. Furthermore, the dataset construction process is only briefly described and does not provide sufficient information regarding annotation protocols and dataset statistics. I recommend that the authors clarify the intended primary contribution and, if REYIA is a key contribution, significantly strengthen its documentation and analysis.
3. Both the REYIA dataset and RLAD-generated data are used jointly to train the segmentation models. How can it be demonstrated that the performance improvement is specifically attributable to the RLAD-generated data or REYIA dataset?
4. RLAD's performance is constrained by the accuracy and generalizability of its pretrained segmentation models (e.g., AV, CD, L). Specifically, using pretrained segmentation models to extract AV, CD, and L from unannotated images may introduce errors, which can subsequently affect the quality of the generated results. The analysis of the impact of the layout extractor in the apppendix is not sufficient.
5. Some aspects of the method are not clearly explained: The random masking strategy is said to apply to all layout components, but the implications of masking AV—especially when generating paired image-segmentation data—are not clearly described. How are image-label pairs constructed when AV is missing? It is also unclear whether the layout components and the fundus image are encoded through the same VAE or separate encoders.

**Strengths Contributions:**

1. The retinal fundus image dataset—REYIA—has been constructed and made public, containing 586 manually segmented images, which greatly promotes research progress in relevant fields.
2. By conditioning the generation on multiple structural components, the method improves anatomical fidelity.
3. The introduction of user input (UI) tokens enables flexible conditional control, allowing partial inputs and thus promoting diversity in generated samples.This approach addresses prior limitations where the diversity of generated AV masks was insufficient, thereby compromising the quality of synthesized images.
4. The experiments comprehensively evaluate the realism of RLAD-generated fundus images, as well as their impact on AV segmentation performance. Evaluations are conducted across In-Domain, Near-Domain, and Out-of-Domain datasets. The method consistently improves performance across different backbone architectures, demonstrating strong cross-domain generalization and architecture-agnostic benefits.
5. Ablation studies further show that incorporating more image centers, conditioning on additional structures (CD and L), and increasing the number of generated samples all lead to performance gains. The effectiveness of RLAD-generated data in scenarios with limited real annotations is also well demonstrated.

---

> ### Author Rebuttal · Authors · 2025-07-29
>
> Thank you for your thoughtful and detailed review. We appreciate your acknowledgement of the contributions, including the public release of the REYIA dataset, the structural conditioning of our generative model, and the consistent improvements across multiple segmentation models and datasets. Your comments helped us clarify the positioning of our work and improve the documentation of both the method and dataset. We have addressed all the concerns you pointed out and will integrate them in the revised camera ready version.
>
> ### 1 & 2. Misalignment Between Submission Track and Core Contribution
>
> While we acknowledge that more elaboration on the novel data contribution is needed, we are confident that our submission aligns with the dataset track through its data-centric contributions. The submission indeed contributes a unique manually annotated dataset (REYIA) as well as a generative model which is used to synthesize additional fundus images given a segmentation mask. We show the value of the two newly contributed data types in significantly and non-incrementally improving AV segmentation performance over state-of-the-art models.
>
> **REYIA Dataset:** We present a collection of 585 manually annotated fundus images with artery/vein (AV) segmentations, sourced from nine diverse datasets across eight different geographical regions. This highlights the variability and diversity of the population samples.
>
> **Synthetic Data Generation Framework:** RLAD generates synthetic fundus images conditioned on real vessel structures. Our experiments demonstrate that incorporating these synthetic images during training can improve segmentation performance. We view this as a data augmentation resource that complements real annotated data. We will add to the kaggle repository the 7230 generated images using RLAD.
>
> In order to better highlight our alignment with the submission track we will add the following paragraphs and associated table to the manuscript:
>
> "The field has benefited from several open AV datasets (TABLE below). While these datasets have been invaluable for the field, challenges remain in terms of scale, diversity of imaging conditions, and geographic representation. REYIA represents the most comprehensive retinal fundus dataset for AV analysis, being the second largest collection in size with 585 images while uniquely combining data from nine diverse sources across multiple geographical location (China, France, Spain, Maghreb, Belgium, USA, Montenegro, and Brazil). Unlike existing single-source datasets, REYIA encompasses both portable and classical fundus imaging modalities, covers a broader range of pathologies (glaucoma, diabetic retinopathy, AMD, and healthy cases), includes the widest variety of field-of-view specifications (30° to ultra-wide) as well as population samples (eight geographical locations). The collection includes resolutions spanning from 829×1531 to 2560×2560 pixels. In addition, the research work contributes 7230 generated synthetic fundus images conditioned on real vessel structures using RLAD.
>
> Most importantly, REYIA ensures annotation quality through a rigorous two-round quality control protocol: initial vessel segmentation by ten junior annotators (ophthalmology students with microvascular research experience) using the Lirot.ai software [1], corrected and validated by a senior medical doctor (ophthalmology resident with PhD specialization in retinal vasculature analysis). Arteriole-venule discrimination was based on established morphological criteria including vessel darkness (venules appearing darker), central light reflex visibility (more prominent in arterioles), vessel caliber differences (venules typically thicker), anatomical positioning patterns near the optic disc (alternating arrangement), and crossing probability rules (same vessel types rarely cross). This systematic quality control and annotation approach, combined with the data diversity makes REYIA a unique resource for real-world AV segmentation model development."
>
> We thank the reviewer for his/her remark and we hope that the additional statement further clarifies the benefit of the new data contributions.
>
> [1] Jonathan Fhima, Jan Van Eijgen, Moti Freiman, Ingeborg Stalmans, and Joachim A Behar. Lirot. ai: a novel platform for crowd-sourcing retinal image segmentations. In 2022 computing in cardiology (CinC), volume 498, pages 1–4. IEEE, 2022. doi: 10.22489/CinC.2022.060.
>
> | Dataset | Geography | # Sources | # Images | FOV (°) | Pathologies | Resolution (px) | Acquisition |
> |---------|-----------|-----------|----------|---------|-------------|-----------------|-------------|
> | DRIVE | Netherland | 1 | 40 | 45 | DR | 584x565 | Fundus image |
> | HRF | Germany | 1 | 45 | 45 | DR, G | 3504x2336 | Fundus image |
> | LES-AV | Belgium | 1 | 20 | 30 | G | 1444x1444 | Fundus image |
> | IOSTAR | Netherland | 1 | 30 | 30 | - | 1024x1024 | Fundus image |
> | AV-WIDE | United States | 1 | 26 | Ultra Wide | - | 829x1531 | Fundus image |
> | UZLF | Belgium | 1 | 240 | 30 | G | 1444x1444 | Fundus image |
> | UNAF | Paraguay | 1 | 15 | 30 | DR | 2056x2124 | Fundus image |
> | RVD | - | 1 | 1270 (video frames) | 30 | - | 1800x1800 | Portable camera |
> | REYIA | Multiple (China, France, Spain, Maghreb, Belgium, USA, Montenegro and Brazil) | 9 | 585 | 30-Ultra wide | G, DR, AMD, H | VAR | Portable and classical fundus image |
>
> *DR = Diabetic retinopathy, G= Glaucoma, H = Healthy, AMD = Age-related Macular Degeneration, VAR = Variable*
>
> ### 3. Attribution of Performance Improvements
>
> We thank the reviewer for his/her remark. Table 3 highlights the improvements achieved using REYA. For example using Swinv2 tiny backbone:
>
> - **Baseline (UZLF only):** 82.1% Local, 75.5% External, 60.6% OOD
> - **With REYIA additions:** 83.1% Local, 79.6% External, 68.9% OOD
> - **Real data improvement:** +1.0% Local, +4.1% External, +8.3% OOD
>
> Similarly, Table 4 shows the supplementary performance achieved using RLAD synthetic data (on Swinv2 tiny backbone):
>
> - **With pretraining:** 83.2% Local, 79.6% External, 69.4% OOD
> - **With RLAD:** 83.3% Local, 79.9% External, 70.8% OOD
> - **Synthetic data improvement:** +0.1% Local, +0.3% External, +1.4% OOD
>
> This separation allows clear attribution of improvements to each component.
>
> In order to clarify the added value of REYA and Synthetic data we will add the following sentence to the discussion:
>
> "When considering the Swinv2 tiny backbone, a performance gain of +4.1% External and +8.3% OOD was obtained when using the newly contributed REYA dataset (Table 3). When adding synthetic data generated using RLAD then an additional +0.3% External, +1.4% OOD in performance were obtained (Table 4). The results highlight the value of the new data contributions (REYA and RLAD generated) on improving AV segmentation."
>
> We thank the reviewer for his/her remark and we hope that the additional statement further clarifies the benefit of the new data contributions.
>
> ### 4. RLAD's Dependency on Pretrained Models
>
> We agree with the reviewer's concern regarding RLAD's dependency on pretrained models for layout extraction and will address this limitation more comprehensively by adding the following paragraph to the discussion section:
>
> "A key limitation of our approach is its reliance on pretrained vessel segmentation models for layout extraction, which introduces a potential bottleneck in the generation pipeline. The quality of synthetic fundus images is inherently constrained by the accuracy of the initial vessel segmentation used as conditioning input. While our experiments in Section F of the supplementary material (Figure 7) illustrate some model tolerance to corrupted inputs and segmentation noise, suggesting robustness to minor extraction errors, we acknowledge that systematic segmentation failures could propagate to the generated images."
>
> ### 5. Method Clarification
>
> We acknowledge the reviewer's request for additional methodological details. We will add the following clarifications to the supplementary material:
>
> "**Training Process:** All layout components (AV, CD, L) are independently masked with some probability, enabling the model to learn both fully and partially conditioned generation. When masked, components are replaced with zero embeddings and UI tokens are updated accordingly.
>
> **Generation Process:** AV is always provided (never masked) to ensure vessel structure preservation, while CD and L can be optionally masked for controlled variation. All components use the same frozen VAE encoder (latent dimension: 4×32×32).
>
> **Paired Data Construction:** The AV mask used for conditioning becomes the ground truth for the generated image, as the generated image inherently contains the vessel structure provided through conditioning."
>
> ### 6. Code Accessibility
>
> We acknowledge the reviewer's request for code accessibility. Complete implementation code, pretrained model weights, and comprehensive documentation are now available at: https://anonymous.4open.science/r/RLAD-45AB/README.md. The repository includes all necessary components for training and inference with RLAD, along with detailed usage instructions.
>
> ---
>
> Thank you again for your constructive feedback. If our revisions have addressed your concerns, we would be grateful if you would consider raising your evaluation. We remain happy to clarify any remaining points.

---

> > ### Comment · Reviewer_UU79 · 2025-08-05
> >
> > All my issues have been addressed. Now I clearly understand the data contributions made by the article. I will raise the rating.

---

### Comment · Area_Chair_UDyb · 2025-08-05

Dear all reviewer,

Could the rebuttal have addressed your concerns? Please join the discussion and update your score.

The AC

---

### Decision · Program_Chairs · 2025-09-18

**Decision:**

Reject

**Comment:**

This paper introduces Retinal Layout-Aware Diffusion (RLAD), a diffusion model for generating synthetic retinal images conditioned on anatomical structures (vessels, optic disc, lesions), and releases RETA, a multi-source dataset of 585 manually segmented retinal images. Key strengths include:
1) RETA’s scale and diversity (9 sources, 8 regions, multiple pathologies), with rigorous annotation validation (Cohen’s κ > 0.98);
2) RLAD’s controllable synthesis, improving segmentation generalization by up to **8.1%** on cross-domain data;
3) Full code/data release (7,230 synthetic images), aligning with DB Track goals.

Weaknesses initially included insufficient dataset benchmarking and ethical documentation, but authors addressed these by:
1) Adding comparisons to existing datasets (e.g., DRIVE, HRF);
2) Including RAI metadata (watermarking, annotator demographics);
3) Fixing accessibility issues (Kaggle links, loading scripts).

During the discussions with the author and reviewers, the reviewers have reached a consensus on acceptance. And finally, we request that all promised revisions and experiments be added to the camera-ready version. The paper delivers a high-impact dataset (RETA) and reproducible augmentation method (RLAD), with sufficient validation and revisions resolving all major concerns.

===== FINAL UPDATE FROM DB Track PCs ====

The final decision for this paper has been taken by the program chairs after consultation with the SACs. All Senior Area Chairs have ranked papers according to the feedback from the AC during the review process. We decided to leave the original meta-review to reflect the opinion of the AC in light of the initial discussions with reviewers and SAC.